# Duodenal Dysbiosis and Relation to the Efficacy of Proton Pump Inhibitors in Functional Dyspepsia

**DOI:** 10.3390/ijms222413609

**Published:** 2021-12-19

**Authors:** Lucas Wauters, Raúl Y. Tito, Matthias Ceulemans, Maarten Lambaerts, Alison Accarie, Leen Rymenans, Chloë Verspecht, Joran Toth, Raf Mols, Patrick Augustijns, Jan Tack, Tim Vanuytsel, Jeroen Raes

**Affiliations:** 1Department of Gastroenterology and Hepatology, University Hospitals Leuven, 3000 Leuven, Belgium; lucas.wauters@kuleuven.be (L.W.); jan.tack@kuleuven.be (J.T.); 2Translational Research in Gastrointestinal Disorders (TARGID), KU Leuven, 3000 Leuven, Belgium; Matthias.ceulemans@kuleuven.be (M.C.); maarten.lambaerts@student.kuleuven.be (M.L.); alison.accarie@kuleuven.be (A.A.); joran.toth@kuleuven.be (J.T.); 3VIB Center for Microbiology, 3000 Leuven, Belgium; raulyhossef.titotadeo@kuleuven.vib.be (R.Y.T.); leen.rymenans@kuleuven.be (L.R.); chloe.verspecht@kuleuven.be (C.V.); 4Department of Microbiology and Immunology, Rega Institute, KU Leuven, 3000 Leuven, Belgium; 5Drug Delivery and Disposition, KU Leuven, 3000 Leuven, Belgium; raf.mols@kuleuven.be (R.M.); patrick.augustijns@kuleuven.be (P.A.)

**Keywords:** duodenum, dysbiosis, proton pump inhibitor, functional dyspepsia

## Abstract

Proton pump inhibitors (PPI) may improve symptoms in functional dyspepsia (FD) through duodenal eosinophil-reducing effects. However, the contribution of the microbiome to FD symptoms and its interaction with PPI remains elusive. Aseptic duodenal brushings and biopsies were performed before and after PPI intake (4 weeks Pantoprazole 40 mg daily, FD-starters and controls) or withdrawal (2 months, FD-stoppers) for 16S-rRNA sequencing. Between- and within-group changes in genera or diversity and associations with symptoms or duodenal factors were analyzed. In total, 30 controls, 28 FD-starters and 19 FD-stoppers were followed. Mucus-associated *Porphyromonas* was lower in FD-starters vs. controls and correlated with symptoms in FD and duodenal eosinophils in both groups, while *Streptococcus* correlated with eosinophils in controls. Although clinical and eosinophil-reducing effects of PPI therapy were unrelated to microbiota changes in FD-starters, increased *Streptococcus* was associated with duodenal PPI effects in controls and remained higher despite withdrawal of long-term PPI therapy in FD-stoppers. Thus, duodenal microbiome analysis demonstrated differential mucus-associated genera, with a potential role of *Porphyromonas* in FD pathophysiology. While beneficial effects of short-term PPI therapy were not associated with microbial changes in FD-starters, increased *Streptococcus* and its association with PPIeffects in controls suggest a role for duodenal dysbiosis after long-term PPI therapy.

## 1. Introduction

Functional dyspepsia (FD) is a common functional gastrointestinal (GI) disorder defined by symptoms originating from the gastroduodenal region [1]. Increasing data point toward duodenal alterations in FD pathophysiology, including mucosal hyperpermeability and low-grade inflammation [2]. The causes are unknown but candidates include luminal acid, bile salts and microbiota [2,3]. Gut commensals play an essential role in nutrient acquisition, colonization resistance, epithelial barrier function and immune development [4,5]. Disruption of the gut ecosystem or dysbiosis has been described in different GI disorders based on stool samples, which do not accurately reflect the mucosal microbiome [6]. Moreover, evidence for dysbiosis is mounting in irritable bowel syndrome (IBS) but is scarce for FD, despite the latter being even more prevalent than IBS [7].

Despite important host–microbiome interactions, the human small intestine is understudied compared to the colon [6,8]. Techniques to characterize the duodenal microbiome have been developed but sample size was small and therapy with proton pump inhibitors (PPI) not taken into account in a previous pilot study in FD [9,10]. Indeed, acid suppression with PPI impacts the gut microbiome, including the fecal and gastric microbiome [11,12]. PPI are the current first-line therapy in FD but long-term efficacy is limited and possibly related to microbiota changes with an increased risk of enteric infections [13]. Deprescribing of PPI has been proposed, especially as FD is not considered an indication for long-term PPI use [14].

Recently, we reported the first prospective evidence for duodenal eosinophil-reducing effects as a therapeutic mechanism of short-term PPI in FD patients [15]. In contrast, duodenal changes were also present and not reversed after withdrawal of long-term PPI in FD, suggesting a role for persistent alterations. In addition, duodenal eosinophil infiltration after PPI was associated with changes in bile salts in healthy controls, which may be related to duodenal dysbiosis [15]. Therefore, the aims of the present study were to (1) characterize the duodenal mucus- and epithelium-associated microbiome of FD patients vs. controls, (2) assess the effect of PPI therapy on the duodenal microbiome and its reversibility after long-term use and (3) study associations with symptoms and duodenal factors. Besides differential mucus-associated genera in FD patients vs. controls, we show an association between *Streptococcus* and duodenal PPI effects in controls, suggesting a role for duodenal dysbiosis after long-term PPI therapy in FD.

## 2. Results

### 2.1. Study Cohort and Sample Overview

In total, PPI therapy was started in 30 controls and 28 FD-starters and withdrawn in 19 FD-stoppers after exclusion of *Helicobacter pylori* (Figure 1). Baseline characteristics were similar between groups, except for the estimated intake of proteins and fiber (Table 1). Median duration of PPI therapy in FD-stoppers was 3.2 years (interquartile range (IQR) 1.5–5.1 years) before withdrawal due to persisting symptoms on long-term PPI. Median (IQR) number of reads was 58,628 (43,510–75,546) for brush and 16,358 (7257–25,931) for biopsy samples (Appendix A). From the 188 brushes and 192 biopsies with >1000 reads, 785 and 762 annotated genera were obtained after sub-setting (for α-diversity) and CLR transformation (for β-diversity and genera abundance), respectively (Appendix A). Microbial load of brush and biopsy samples was similar between groups (Appendix A).

### 2.2. Duodenal Microbiome Is Altered in FD Patients with PPI Effects

We first assessed the relative importance of all variables explaining the duodenal mucus- (brush) and epithelium-associated (biopsy) microbial variation (community-wide shifts). Significant inter-individual variation (subject) with limited group- and only-PPI effects for brushes were found (Table 2). In multivariate models, subject had a contribution of 16.01% (*p*_adj_ = 0.002) with PPI adding to its contribution (*p*_adj_ = 0.01) to reach a total explanatory power of 16.75% for brush samples, while only a significant and smaller contribution of subject was found for biopsy samples (R^2^ = 5.81%, *p*_adj_ = 0.002). Despite the significant effect of the sampling location (brush or biopsy) when combining all samples (*n* = 380), biopsies were more likely contaminated (Appendix A). Besides the clustering of brush and biopsy samples, a significant but smaller effect of group was found for all samples with an effect of PPI for brush but not biopsy samples (Appendix A). No association of other host factors, including duodenal eosinophils and mast cells, or dietary intake with community variation was found using univariate dbRDA.

### 2.3. Specific Effects on Genera and Diversity after Short-Term PPI

Despite the absence of major shifts in community composition, between-group differences in specific genera were found, including lower abundances of mucus-associated *Neisseria* (FDR < 0.001), *Porphyromonas* (FDR = 0.003), *Selenomonas* (FDR = 0.02), *Haemophilus* (FDR = 0.03) and *Fusobacterium* (FDR = 0.06) in FD-starters vs. controls (Figure 2A) and a decrease in *Prevotella* (FDR = 0.03) after PPI. The lower *Neisseria* (β = −1.48 ± 0.59, *p* = 0.02) and *Porphyromonas* (β = −2.17 ± 0.65, *p* = 0.001) abundance was confirmed in FD-starters vs. controls at baseline (off-PPI) using mixed models, with decreased *Porphyromonas* (β = −1.34 ± 0.56, *p* = 0.02) in controls after PPI (Figure 2B,C and Appendix A). In addition, *Prevotella* decreased in controls (β = −0.92 ± 0.43, *p* = 0.03) and FD-starters (β = −1.65 ± 0.47, *p* < 0.001) after PPI (Figure 2D). Based on the findings of a recent study [16], we analyzed mucus-associated *Streptococcus* abundance, which was similar between groups but increased after PPI in controls (β = 0.31 ± 0.12, *p* = 0.01) and FD-starters (β = 0.22 ± 0.13, *p* = 0.03) (Figure 2E). In contrast, no differentially abundant epithelium-associated genera were found (Appendix A).

Mucus-associated richness was lower in FD-starters vs. controls (β = −0.85 ± 0.39, *p* = 0.03) at baseline, with a significant decrease in controls (β = −0.73 ± 0.32, *p* = 0.03) after PPI (Table 3). Shannon’s and Simpson’s index were similar between groups with a decrease in both controls (β = −0.31 ± 0.11, *p* = 0.008 and β = −1.06 ± 0.38, *p* = 0.007) and FD-starters (β = −0.39 ± 0.13, *p* = 0.003 and β = −1.26 ± 0.41, *p* = 0.003, respectively) after PPI (Table 3). No significant changes were observed in epithelium-associated α-diversity. Finally, significant spatial (Appendix A and Appendix A) but not temporal variation (Appendix A) of the duodenal mucus- and epithelium-associated microbiome was found (Appendix A). Thus, baseline differences and duodenal effects of PPI therapy were only found for specific mucus-associated genera and diversity, with stable mucus- and epithelium-associated bacterial communities in the absence of PPI.

### 2.4. Persisting Microbiota Alterations after Withdrawal of Long-Term PPI

In FD-stoppers, mucus-associated *Neisseria* abundance was higher vs. FD-starters (FDR = 0.09) but not controls. Higher *Neisseria* abundance was confirmed in FD-stoppers vs. FD-starters on-PPI (β = 1.41 ± 0.69, *p* = 0.04) using linear mixed models, pointing to differences between short- and long-term use of PPI in FD. *Streptococcus* abundance was similar in FD-stoppers vs. controls on-PPI but higher off-PPI (β = 0.31 ± 0.16, *p* = 0.03), suggesting persistent microbial alterations with no changes after PPI withdrawal (*p* > 0.05). In contrast, a decreased abundance of *Rothia* (FDR = 0.01) and *Stomatobaculum* (FDR = 0.09) and increased *Prevotella* (β = 1.21 ± 0.55, *p* = 0.03) were found after PPI withdrawal. Besides differentially abundant epithelium-associated genera including *Dyella*, richness was lower in FD-stoppers vs. FD-starters (both *p* = 0.03) and controls (both *p* < 0.01) off-PPI but with no effect of PPI withdrawal (Appendix A). No changes were observed in mucus-associated α-diversity (Appendix A). These findings indicate persisting mucus- and epithelium-associated microbial alterations despite withdrawal of long-term exposure to PPI in FD patients.

### 2.5. Duodenal Dysbiosis in Relation to Efficacy of PPI in FD Patients and Controls

Based on our findings of mucus-associated microbial alterations in FD-starters vs. controls, correlations with symptoms and duodenal eosinophils were assessed. Baseline abundance of *Porphyromonas* correlated with symptoms (r = −0.35) and eosinophils (r = −0.43) (Figure 3A,B) and *Neisseria* with symptoms (r = −0.33, all FDR = 0.04) in controls and FD-starters. No correlations were found with other host factors, including duodenal mast cells, or dietary intake. Next, we addressed whether symptom- or eosinophil-reducing effects of PPI therapy were associated with microbial changes, as clinical efficacy was only found in FD patients with an average or greater decrease in eosinophils [15]. However, the reduction in symptoms or eosinophils after PPI was similar for different levels of the standardized (relative to the mean) changes in *Porphyromonas* (Figure 3C,D), *Neisseria*, *Prevotella* and *Streptococcus* (Appendix A). Although baseline diversity correlated with symptoms (r = −0.57, FDR = 0.02), clinical efficacy of PPI was also not associated with changes in diversity in FD-starters (Appendix A).

In contrast, increased duodenal eosinophils were found in controls after PPI and associated with duodenal bile salts, suggesting a role of luminal changes [15]. Duodenal eosinophils correlated with *Porphyromonas* (r = −0.44, FDR = 0.04) and *Streptococcus* (r = 0.4, FDR = 0.06) in controls. Moreover, increased duodenal eosinophils after PPI were associated with changes in *Streptococcus* (Figure 3E) but not *Porphyromonas* (Appendix A). Interestingly, increased duodenal secondary bile salt concentrations were also associated with changes in *Streptococcus* after PPI therapy (Figure 3F), while an inverse association of *Streptococcus* with changes in secondary bile salts was not found. Thus, duodenal PPI effects were associated with microbiota changes in controls and not FD-starters, with a potential role for *Streptococcus* in determining duodenal eosinophilia during PPI therapy.

## 3. Discussion

Despite the high prevalence, the pathophysiology of FD is incompletely understood. Duodenal alterations have been reported in different studies, but the presence and potential role of duodenal dysbiosis are still unclear. Therefore, we studied the mucus- and epithelium-associated duodenal microbiome, including temporal variation (off-PPI) and changes after routine PPI-therapy (FD-starters and controls) and withdrawal of long-term PPI (FD-stoppers). Significant inter-individual variation was found with limited group- and only-PPI effects for brushes. Interestingly, specific mucus-associated genera differed between FD patients and controls, including *Neisseria* and *Porphyromonas*, which were less abundant in FD and correlated with symptoms and duodenal eosinophils. *Streptococcus* increased and *Prevotella* and diversity decreased after short-term PPI therapy. Although symptom- and eosinophil-reducing effects of PPI were not associated with microbial changes in FD, increased duodenal eosinophils and bile salts after PPI were associated with changes in *Streptococcus* in controls. As duodenal PPI effects were associated with microbiota changes in controls, these data also suggest similar effects with limited reversibility after long-term PPI in FD patients (Figure 4).

Inter-individual variation of the duodenal microbiome was significant but with no temporal variation in the absence of PPI, suggesting that bacterial adherence to mucus and epithelium may indeed be important for persistent colonization [6]. The lack of group effects in multivariate models was not unexpected as the majority of patients with functional GI disorders had a “healthy-like” microbial composition with decreased α-diversity, *Porphyromonas*, *Neisseria*, *Haemophilus* and *Fusobacterium* vs. controls in a previous study [17]. Duodenal *Porphyromonas* abundance was also lower in IBS patients and the baseline difference and correlations with symptoms and duodenal eosinophils in our cohort point to a potential role of this genus in FD pathophysiology [18]. No correlations were found with other (host) factors, including duodenal mast cells. While we observed a PPI-induced increase in *Streptococcus* and decrease in *Prevotella* and diversity in FD-starters and controls, only *Prevotella* was associated with changes in duodenal pH. Although pH effects are attenuated by the distal duodenum, bacterial and direct targets of PPI could also contribute to changes in specific genera, including *Streptococcus* [11,12].

We showed that eosinophil-reducing and not acid-suppressive or barrier-protective effects of short-term PPI-therapy were associated with clinical efficacy in FD patients [15]. While symptom- and eosinophil-reducing effects of PPI were not associated with microbial changes in FD-starters, increased duodenal eosinophils and bile salts after PPI were associated with changes in *Streptococcus* in controls. Recently, GI eosinophil infiltration in the presence of an offending antigen was found to be transient, suggesting potential homeostatic roles of eosinophils [19]. Although anti-eosinophil effects of PPI are known, acid suppression was also linked to Th2-type reactions in mechanistic and population-based studies, suggesting a potential role of PPI-induced dysbiosis [20]. Although duodenal dysbiosis could be an epiphenomenon of PPI, the association between secondary bile salts and *Streptococcus* (and not vice versa) rather points to causal effects. Interestingly, gastric *Streptococcus* abundance was also higher with long-term PPI use and potentially linked to persisting symptoms in FD patients [12]. Despite a potential role for probiotics, evidence for clinical efficacy and duodenal effects are lacking in FD [21,22].

This study was performed at a single and tertiary care center, which may limit the generalizability of our findings although characteristics were comparable to the general FD-population [23]. Despite different procedural and analytical preventive measures, contamination could still influence the results as statistical removal of potential contaminants is not intended to detect cross-contamination [24]. Moreover, cross-contamination with true signals would also preclude manual removal of sequences present in negative controls and subanalyses with 10,000 reads illustrated that richness was falsely inflated using a lower cut-off, which was not the case for Shannon’s and Simpson’s index [25]. Because of the risk of contamination with biopsies, results need to be interpreted with caution. We studied bacterial composition and not function (meta-genomics) or other micro-organisms. Moreover, baseline correlations and associations with symptoms and duodenal factors do not prove causality.

Strengths of this prospective study include the homogenous patient and control populations with repetitive sampling of the duodenal mucus- and epithelium-associated microbiome, which have not yet been compared. Besides the interventional design with PPI therapy, temporal variation was also studied in the absence of PPI. Methodological optimizations were performed for sampling and storage procedures, as contamination may arise from PBS or RNA-later solutions, and we included more than the suggested number of negative controls for the detection of contaminants [24]. Potential effects of diet were measured but not expected as duodenal samples were taken in a fasted state, with limited substrate availability compared to the colon. Although a longer duration of PPI intake and withdrawal would be needed to study the potential role and reversibility of microbial changes in FD patients, our results also illustrate the limitations of cross-sectional studies of the duodenal microbiome.

## 4. Materials and Methods

### 4.1. Study Population

Two interventional studies aimed at characterizing the duodenal microbiome and effect of PPI were conducted over 2 years (April 2018–April 2020) at a single center, according to the Declaration of Helsinki and Good Clinical Practice regulations after approval by the Ethics Committee of the University Hospitals Leuven (numbers S60953/S60984). The clinical and duodenal mucosal data of both studies have been reported before [15]. The primary analysis on the duodenal microbiome is presented here, also in relation to PPI therapy. Symptomatic FD patients, diagnosed according to Rome IV criteria [1], were included if they had not been treated with PPI therapy (4 weeks healing dose) or other acid suppression <3 months before inclusion (“FD-starters”), or if persistent symptoms after >1 month of at least one daily dose of PPI (“FD-stoppers”). Symptom severity was assessed using the patient assessment of upper GI symptom severity index (PAGI-SYM), ranging from 0 (none) to 5 (very severe) over a two-week recall period. Age- and gender-matched healthy controls without GI symptoms were also recruited. All subjects were aged 18 to 64 years old, with no active psychiatric, atopic, inflammatory or metabolic conditions. Use of immunosuppressants, anti- or probiotics <3 months were exclusionary. Written informed consent was obtained from all subjects before inclusion.

### 4.2. Sample Collection

The study design is shown in Figure 1. During upper GI endoscopy, aseptic biopsies from the second portion of the duodenum (D2) were collected using the sheathed and sealed Brisbane Aseptic Biopsy Device (BABD) (MTW, Wesel, Germany) [9], with additional precautions to avoid contamination. Next, a sterile brush (Zhuji Pengtian Medical Instrument Co., Zhejiang, China) was advanced while leaving the sheathed BABD in place for brushing, on the opposite side from where the biopsy sample was taken. Aseptic procedures were repeated after 2–4 weeks to assess the variability of the duodenal microbiota (off-PPI) and after an additional 4 weeks of routine PPI therapy (Pantoprazole 40 mg once daily) in controls and FD-starters (on-PPI). For FD-stoppers, all procedures were performed at baseline (on-PPI) and after 8 weeks of PPI withdrawal (off-PPI). Routine gastric (for *Helicobacter pylori*) and duodenal biopsies (for histology) and fluids (pH and bile salts) were collected in all subjects at baseline and follow-up (Appendix A).

### 4.3. Sample and Data Processing

Aseptic duodenal brushes (mucus-associated) and biopsies (epithelium-associated microbiota) were transferred in sterile, nuclease-free tubes using sterile needles and wire cutters. Samples were immediately snap-frozen and stored at −80 °C. All procedures were performed under sterile conditions in a biohazard type II cabinet using cleaned (RNase AWAY, Molecular BioProducts, San Diego, CA, USA) and UV-irradiated equipment [26]. DNA was extracted using the AllPrep^®^ DNA/RNA Mini kit (Qiagen, Hilden, Germany) according to the manufacturer’s instructions, with the addition of 1 extraction blank for every 5 samples (random order for brushes and biopsies) [27]. Bacterial DNA quantification (Uni16S) was performed before amplification of the 16S rRNA V4 hypervariable region as previously described (Appendix A) [26,28]. Final DNA concentration and fragment lengths were determined before equimolar pooling and dual-index sequencing using the Illumina MiSeq platform, yielding paired-end reads of 250 bases length in each direction. Quality control and annotation of 16S rRNA-sequences was followed by removal of potential contaminants using decontam (prevalence-based method, threshold 0.5) [24], before further analysis at genus level using a minimum of 1000 reads for all samples.

### 4.4. Statistical Analysis

The primary analysis was the duodenal microbiome composition in FD patients vs. controls for both locations (brush and biopsy). The association of subject, group, treatment and demographics to bacterial community variation was studied using distance-based redundancy analysis (dbRDA, genus-level Aitchison distance) after centered log-ratio (CLR) transformation. Clustering of significant variables was also determined on principal component analyses (permutational MANOVA). Differential genera abundance was assessed using (un-)paired *t*-tests with correction for multiple testing (Benjamini–Hochberg FDR < 0.1). Next, genera of interest, richness (Observed, Chao1) and diversity (Shannon, Simpson) metrics were compared between and within groups using linear mixed models with group (controls, FD-starters or FD-stoppers) as between- and treatment (off- or on-PPI) as within-subject factors, including their interaction. In addition, spatial and temporal variation were assessed with, respectively, location (brush or biopsy) or visit (baseline or variability off-PPI) as within-subject factors in controls and FD-starters. Finally, Spearman correlations (FDR < 0.1) and associations between PPI-induced changes in symptoms or duodenal factors and duodenal microbial variables were determined. All analyses were performed using R v4.0.3 (R Studio, Boston, MA) and SAS software v9.4 (SAS Institute, Cary, NC) and two-tailed *p*-values < 0.05 were considered significant unless otherwise specified. For mixed models, least squares means estimates (β) are given as mean ± standard error. Details on microbiota and statistical analyses are given in the Appendix A.

## 5. Conclusions

In conclusion, we showed significant inter-individual variation of the duodenal mucus- and epithelium-associated microbiome, which was stable over time in the absence of PPI. Specific changes were only found for the mucus-associated microbiota, with a potential role of *Porphyromonas* and increased *Streptococcus* abundance. While symptom- and eosinophil-reducing effects of short-term PPI therapy were not associated with microbial changes in FD, the magnitude of the increased *Streptococcus* was associated with duodenal PPI effects in controls. Indeed, persistently increased *Streptococcus* may also cause similar alterations after long-term PPI exposure in FD. Whether this could be prevented or treated with microbiota-directed treatments should be further studied, especially in FD patients with persisting symptoms on first-line therapy.

## Figures and Tables

**Figure 1 ijms-22-13609-f001:**
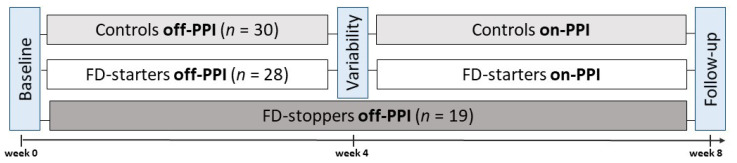
Study design and procedures. Procedures were performed at baseline, after 2–4 weeks (variability) and after 4 weeks of Pantoprazole 40 mg once daily (follow-up) in controls and FD-starters. For FD-stoppers, procedures were performed at baseline and after 8 weeks of PPI withdrawal (off-PPI).

**Figure 2 ijms-22-13609-f002:**
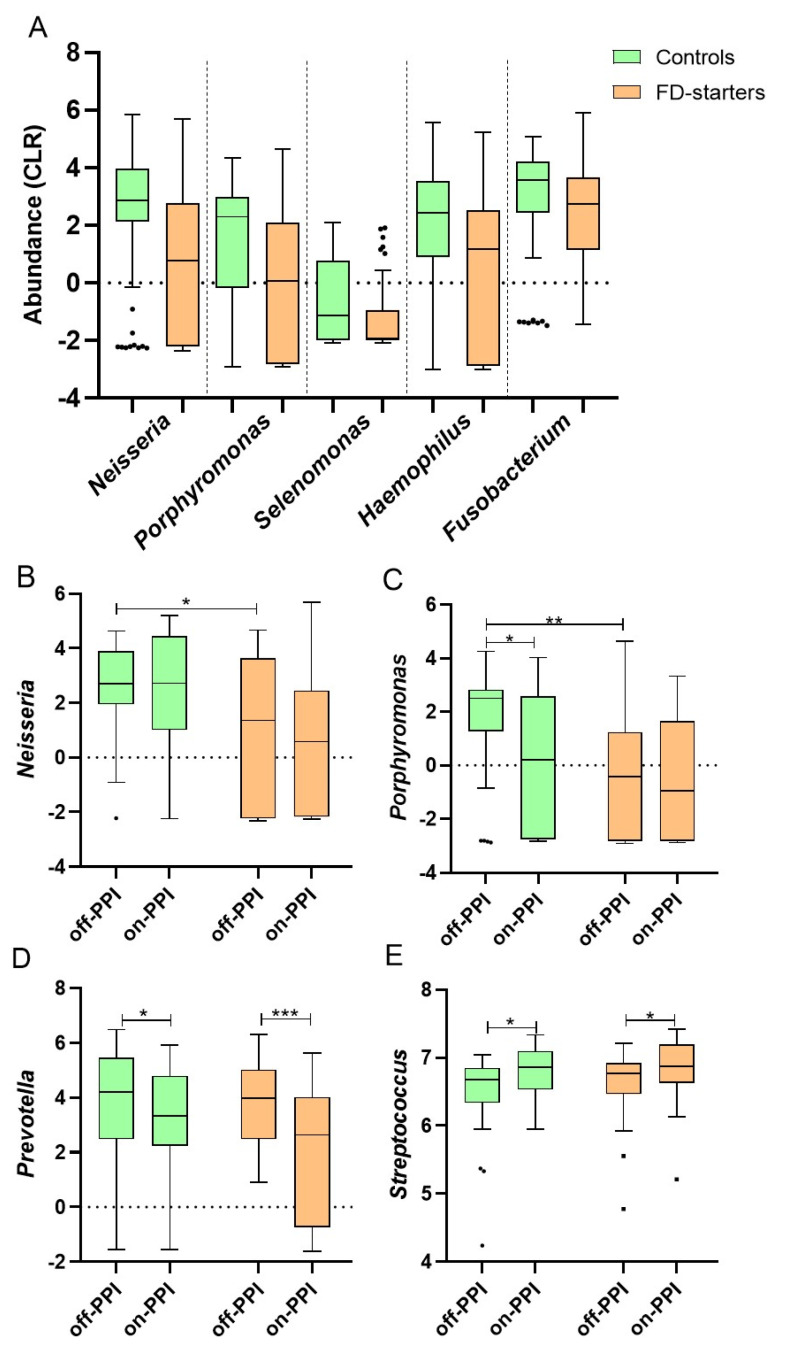
Relative abundances of mucus-associated duodenal genera across groups and PPI status. (**A**) Differential genera abundance for brush samples of FD-starters vs. controls; (**B**) changes in *Neisseria,* (**C**) *Porphyromonas,* (**D**) *Prevotella* and (**E**) *Streptococcus* according to group and PPI status. Tukey boxplots of CLR-transformed genera with median, IQR and 1.5 * IQR whiskers (outliers beyond). (**A**) FDR < 0.1 for all genera (between groups). (**B**–**E**) * *p* < 0.05, ** *p* < 0.01, *** *p* < 0.001.

**Figure 3 ijms-22-13609-f003:**
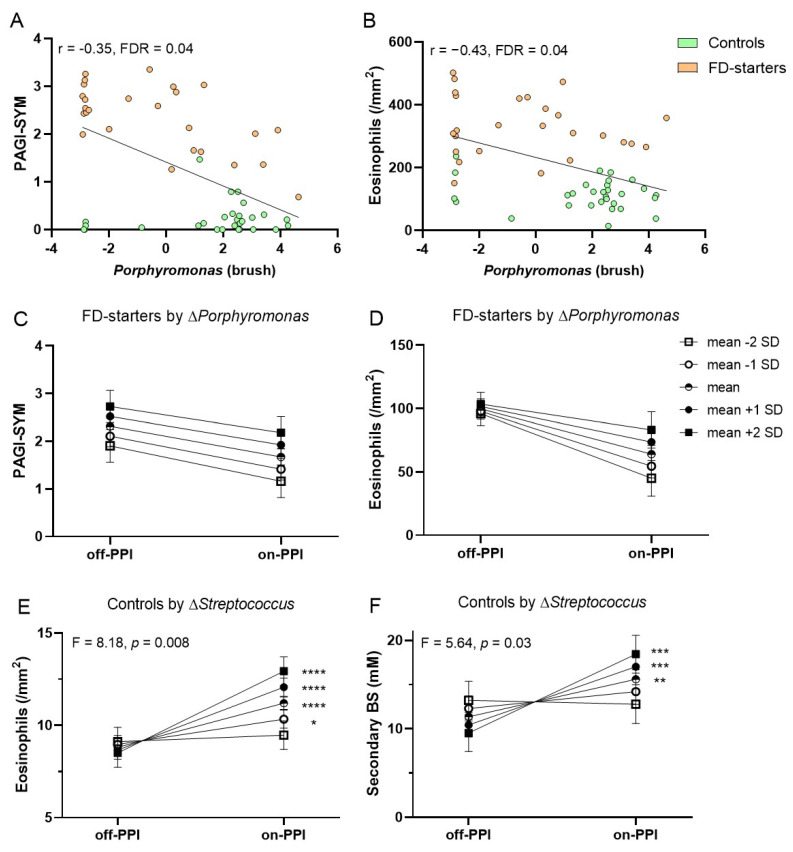
Correlations and associations between PPI-induced changes in symptoms or duodenal factors and duodenal microbial variables. (**A**,**B**) Correlation between mucus-associated *Porphyromonas* and symptoms (**A**) or duodenal eosinophils (**B**) in controls and FD-starters (off-PPI). (**C**,**D**) Association between clinical (**C**) and eosinophil-reducing (**D**) effects of PPI with changes in *Porphyromonas* in FD-starters. (**E**,**F**) Association between duodenal eosinophil infiltration (**E**) and secondary bile salts (BS) (**F**) after PPI with changes in *Streptococcus* in controls. (**C**–**F**) evolution in symptoms and (Box–Cox transformed) duodenal eosinophils or secondary BS by changes in mucus-associated genera, where mean corresponds to an average change (Δ = 0) and mean +/− 1 or 2 SD to an above or below average change. * *p* < 0.05, ** *p* < 0.01, *** *p* < 0.001, **** *p* < 0.0001.

**Figure 4 ijms-22-13609-f004:**
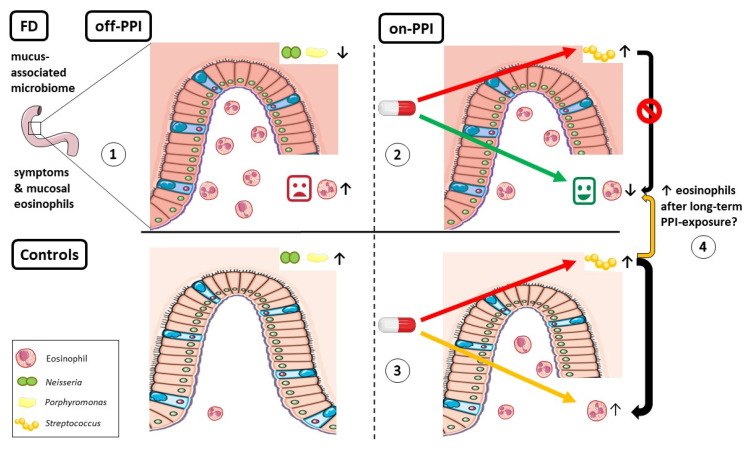
Graphical summary. (**1**) Mucus-associated *Neisseria* and *Porphyromonas* were less abundant in FD vs. controls and correlated with symptoms and duodenal eosinophils. (**2**) Microbial changes, including increased *Streptococcus*, were not associated with beneficial symptom- and eosinophil-reducing effects of short-term PPI therapy in FD. (**3**) In contrast, increased *Streptococcus* was associated with duodenal eosinophil infiltration after PPI in controls. (**4**) Persistently higher *Streptococcus* abundance suggested a role for similar duodenal PPI effects in FD patients after long-term PPI.

**Table 1 ijms-22-13609-t001:** Baseline characteristics of healthy controls, FD-starters (off-PPI) and FD-stoppers (on-PPI).

Group	Healthy Controls (*n* = 30)	FD-Starters (*n* = 28)	FD-Stoppers (*n* = 19)	*p*-Value
Demographic:				
Age (years)	27 (24–33.5)	27 (23.5–34.5)	32 (26.8–49.5)	0.18
Female (%)	21 (70)	24 (86)	14 (74)	0.35
BMI (kg/m^2^)	23 (20–25.3)	22 (19–24)	21.5 (20.8–24.3)	0.56
FD subtypes:				
PDS subtype (%)	NA	15 (54)	10 (53)	0.95
EPS subtype (%)	NA	3 (11)	6 (32)	0.07
Overlap (%)	NA	10 (35)	3 (15)	0.13
Daily food intake:				
Energy (kcal/day)	1419 (1308–1627)	1186 (974.9–1621)	1284 (937.7–1617)	0.35
Sugars (g/day)	175.7 (148.2–187.5)	148.3 (115.1–222.5)	143.9 (93.42–194.4)	0.25
Fat (g/day)	46.87 (41.7–57.1)	40.8 (34.4–52.5)	45.7 (31.2–61.5)	0.51
Fiber (g/day)	18.4 (15.2–21.5)	15.9 (9.3–23)	12.8 (8.3–17.4) *	0.02
Protein (g/day)	175.7 (148.2–187.5)	148.3 (115.1–222.5) *	143.9 (93.4–194.4) *	<0.01

* *p*_adj_ < 0.05 vs. controls (post hoc Dunn tests) after Kruskal–Wallis test with Chi2 = 7.58 (fiber) and 10.22 (protein).

**Table 2 ijms-22-13609-t002:** Duodenal mucus- and epithelium-associated bacterial community variation.

Univariate dbRDA	Mucus-Associated (Brush)				Epithelium-Associated (Biopsy)			
	F-Value	R^2^ (%)	*p*-Value	*p*_adj_-Value	F-Value	R^2^ (%)	*p*-Value	*p*_adj_-Value
subject	1.47	16	0.001	0.006	1.16	5.4	0.002	0.01
group	2.1	1.16	0.002	0.006	1.52	0.51	0.03	0.09
PPI	1.64	0.34	<0.05	0.09	0.92	0	0.41	0.51
gender	1.02	0.01	0.38	0.38	1.05	0.03	0.4	0.4
age	1.16	0.09	0.23	0.28	1.47	0.25	0.06	0.12
BMI	1.52	0.28	0.07	0.1	1.05	0.03	0.32	0.4

Univariate distance-based redundancy analysis with individual effect sizes of subject (inter-individual variation), group, treatment and demographics, assuming covariate independence. Variables remaining significant after adjustment for multiplicity (Benjamini–Hochberg) were entered in a stepwise multivariate model for the mucus- and epithelium-associated microbiome variation.

**Table 3 ijms-22-13609-t003:** Duodenal α-diversity metrics in controls and FD-starters before and after PPI therapy.

Group	Controls		FD-Starters		*p*_adj_ Value
Treatment	Off-PPI (*n* = 30)	On-PPI (*n* = 30)	Off-PPI (*n* = 28)	Off-PPI (*n* = 30)	
Brush					
Observed	42.03 ± 1.36	37.79 ± 1.9 *	36.81 ± 1.6	34.79 ± 1.67	1
Chao1	48.98 ± 1.91	44.81 ± 2.71	43.24 ± 2.28	44.89 ± 2.75	0.62
Shannon	2.32 ± 0.05	2.05 ± 0.08 **	2.24 ± 0.07	1.92 ± 0.1 **	0.66
Simpson	0.79 ± 0.01	0.72 ± 0.02 **	0.78 ± 0.02	0.68 ± 0.03 **	0.71
Biopsy					
Observed	43.07 ± 3.33	41.43 ± 6.12	43.36 ± 7.08	42.37 ± 7.32	0.97
Chao1	45.43 ± 3.82	43.58 ± 6.95	47.74 ± 8.62	45.36 ± 8.53	0.99
Shannon	2.82 ± 0.07	2.8 ± 0.1	2.69 ± 0.12	2.69 ± 0.12	1
Simpson	0.88 ± 0.01	0.89 ± 0.01	0.86 ± 0.02	0.87 ± 0.02	1

* *p* < 0.05, ** *p* < 0.01 (within group) with decreased mucus-associated diversity after PPI, which was similar between groups (*p*_adj_).

## Data Availability

The authors confirm that the data supporting the findings of this study are available within the article.

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
