# Peer review of "Duodenal Dysbiosis and Relation to the Efficacy of Proton Pump Inhibitors in Functional Dyspepsia"

_ijms, 2021, doi:10.3390/ijms222413609_

Round 1

Reviewer 1 Report

This manuscript is demonstrated the duodenal dysbiosis and relation to the efficacy pf PPI in FD patients. The point of view and the results are interesting. There are a few issues that should be addressed prior to the publication in IJMS.  

  1. FD is diagnosed according to the subjective symptoms, and it is important to assess the severity of symptoms. The detailed methods of symptom assessment in FD patients should be described in materials and methods.
  2. How is the status of Helicobacter ,pylori infection in each FD patient? In addition, it is important whether the eradication therapy to pylori is performed or not in the case of H.pylori positive patients. These points should be described if possible.
  3. The reason why withdrawal of long-term PPI therapy in FD-stoppers should be written in baseline characteristics.

Author Response

We thank the reviewer for the constructive comments, please find our responses below (point-by-point):

1. The PAGI-SYM questionnaire was collected at each visit (see supplementary methods) and the range for the total score was repeated in the Materials and Methods to avoid potential confusion.

2. We confirm that Helicobacter pylori was excluded using gastric biopsies in all subjects (see supplementary methods) and we have added this information in both the Results and Materials and Methods to avoid potential confusion.

3. We thank the reviewer and have clarified the withdrawal due to refractory symptoms on long-term PPI in the Results.

Reviewer 2 Report

  1. Please present all scientific name of the bacteria in this manuscript.
  2. Please add desxription of Figure 3 legend for ****................p < .0001? 
  3. Please add other cells' description and effects in addition to the description of eosinophils in this manuscript.

Author Response

We thank the reviewer for the constructive comments, please find our responses below (point-by-point):

1. We thank the reviewer for this comment. For our conservative analysis, only genera names are reported due to the low resolution of the amplicon sequencing data for some species belonging to specific genera. The species name for the top amplicon sequence variant (ASV) of important genera are given below:

Streptococcus match 100% with Streptococcus pseudopneumoniae and Streptococcus oralis subsp. Tigurinus

Porphyromonas match 100% with Porphyromonas pasteri

Selenomonas match 100% with Selenomonas sputigena and Variovorax paradoxus

Fusobacterium match 100% with Fusobacterium pseudoperiodonticum and Fusobacterium periodonticum

Prevotella match 100% with Prevotella melaninogenica

Neisseria match 100% with Neisseria subflava and Neisseria perflava

2. We apologize for the omission and have done this accordingly in Figure 3.

3. Thank you for this comment. We have clarified that the effect of other host factors on community variation using univariate dbRDA included duodenal eosinophils and mast cells (see Results, section 2.2).

Moreover, we also clarify that no correlations were found between microbial alterations with other host factors including duodenal mast cells (see Results, section 2.5).

This important comment was also added to the Discussion (page 8, line 216) to avoid potential confusion.